

# Measurement Report: Vertically resolved Atmospheric Properties Observed over the Southern Great Plains with Uncrewed Aerial System - ArcticShark

Fan Mei[1], Qi Zhang[2], Damao Zhang[1], Jerome D. Fast[1], Gourihar Kulkarni[1], Mikhail S. Pekour[1], Christopher R. Niedek[2], Susanne Glienke[1], Israel Silber[1], Beat Schmid[1], Jason M. Tomlinson[1], Hardeep S. Mehta[3], Xena Mansoura[3], Zezhen Cheng[3], Gregory W. Vandergrift[3], Nurun Nahar Lata[3], Swarup China[3], Zihua, Zhu[3]

[1]Atmospheric, Climate, and Earth Sciences, Pacific Northwest National Laboratory, Richland, WA, 99352, USA
[2]Department of Environmental Toxicology, University of California, Davis, 95616, USA
[3]Environmental Molecular Sciences Laboratory, Pacific Northwest National Laboratory, Richland, WA, 99352, USA

*Correspondence to*: Fan Mei (fan.mei@pnnl.gov)

**Abstract.** This study presents the unique capability of the DOE ArcticShark – a mid-size Uncrewed Aerial System (UAS) – for measuring vertically resolved atmospheric properties over the Southern Great Plains (SGP) of the United States. Focusing on atmospheric states and aerosol properties, we overview measurements from 32 research flights (~ 97 flight hours) carried out in 2023. Our data from March, June, and August 2023 reveal distinctive seasonal patterns within the atmospheric column through unique chemical composition measurements. These two measurement techniques— in situ and remote sensing—provide valuable insights into their consistency and complementarity. The August operations, aided by a visual observer on a chase plane, allowed for extensive UAS coverage, surpassing typical UAS operation envelopes. Furthermore, we demonstrate the capabilities of the ArcticShark through several case studies, including the analyses of correlations between UAS-derived atmospheric profiles and conventional radiosonde measurements, as well as the derivation of vertically resolved profiles of aerosol chemical, optical, and microphysical properties. These case studies highlight the versatility of the ArcticShark UAS as a powerful tool for comprehensive atmospheric research, effectively bridging data gaps and enhancing our understanding of vertical atmospheric structures in the region.

## 1 Introduction

The Southern Great Plains (SGP) region of the United States has long been a focal point for atmospheric research due to its unique geographical and meteorological characteristics (Phillips and Klein, 2014; Williams et al., 2016). Extending across several states, including Oklahoma, Kansas, and Texas, this area offers diverse environmental conditions, making it an ideal location for studying various atmospheric phenomena (Sisterson et al., 2016; Song et al., 2005). This region is also susceptible to extreme weather events (Kelley and Ardon-Dryer, 2021; Mullens and McPherson, 2019). All of these factors led the Department of Energy (DOE) Atmospheric Radiation Measurement (ARM) Program to establish its first comprehensive



measurement site at this location in the 1990s (Sisterson et al., 2016). For 30 years, measurement capabilities at the ARM SGP observatory have kept expanding, including multiple observational platforms with comprehensive instruments for extensive atmospheric, aerosol, and cloud observations. Researchers have utilized the long-term observations from the ARM SGP observatory to gain valuable insights into the dynamics of convective systems, to enable the development of more accurate

climate model simulation, and to further investigate aerosol-cloud interactions (Phillips et al., 2017; Tao et al., 2019; Zheng et al., 2020).

Moreover, the ARM SGP has been a hub for pioneering efforts in atmospheric remote sensing to provide a vertical context of atmospheric processes. Radiosondes are launched regularly to collect temperature, humidity, and pressure data at various altitudes (Berg et al., 2015; Gartzke et al., 2017). State-of-the-art instruments, such as radar systems, lidars, and

advanced meteorological towers, have been deployed to capture data on the vertical structure and dynamics of the atmosphere (Dupont et al., 2011; Thorsen and Fu, 2015; Turner et al., 2016; Jensen et al., 2016; Naud et al., 2003). These capabilities have revolutionized ARM's ability to monitor and analyze atmospheric processes, from boundary layer evolution to cloud microphysics (Dupont et al., 2011; Kennedy et al., 2014; Ou et al., 2002; Riedi et al., 2001; Zhang et al., 2013). Although the continuous monitoring of boundary layer dynamics, along with specific aerosol and cloud vertical properties provided by

radiosondes and remote sensing measurements, provides valuable data, these methods have certain limitations, such as reduced vertical measurement accuracy due to dense clouds and heavy aerosol pollution and insufficient spatial and temporal resolution (Balsamo et al., 2018; Geerts et al., 2018; Rahman, 2023).

Airborne measurements offer crucial insights into the dynamic interactions within Earth's atmosphere due to their extensive spatial coverage, high vertical resolution, and flexibility (Wendisch and Brenguier, 2013). In the past decades, the

SGP observatory has functioned as a central hub, facilitating numerous field studies for collaborative research involving ground and airborne measurements (Andrews et al., 2004; Delle Monache et al., 2004; Feingold et al., 2006; Knobelspiesse et al., 2008; Vogelmann et al., 2012; Biraud et al., 2013; Turner et al., 2014; Endo et al., 2015; Lu et al., 2016; Fast et al., 2019; Schobesberger et al., 2023). During these field campaigns, research aircraft were deployed to conduct intensive observations. The airborne platforms carried specialized instruments at various altitudes to capture detailed information on atmospheric

properties in the SGP region, such as seasonal differences in the vertical profiles of aerosol optical properties (Andrews et al., 2011). The presence of varied land cover, including agricultural fields, grasslands, and urban areas, also offers an excellent opportunity for examining land-atmosphere interactions and understanding how different surfaces influence local weather patterns, energy fluxes, and greenhouse gas exchanges (Fast et al., 2022; Fast et al., 2019; Parworth et al., 2015; Tao et al., 2019; Wang et al., 2023; Zheng et al., 2020).

Anchored at the SGP observatory, the ARM program has continually expanded its capabilities by developing various observational platforms to support the science community. To improve the current understanding of cloud-aerosol interactions, radiative processes, and the impacts of aerosols on both regional and global climate, the ARM program has enhanced its capabilities by incorporating tethered balloon systems and UAS alongside traditional (crewed) aircraft since 2017 (Creamean et al., 2021; Dexheimer et al., 2019; Mei et al., 2022). The ARM Aerial Facility (AAF) has successfully transitioned a mid-



size UAS – the ArcticShark, from test flights to an operational platform available to community users (https://arm.gov/news/facility/post/97628). The ArcticShark offers flexibility, cost-effectiveness and operational advantages. It is highly suitable for supporting the DOE mission to enhance our understanding of atmospheric processes and enable more precise and comprehensive environmental monitoring.

     This paper introduces a novel dataset of airborne measurements collected in 2023 above the central facility of the ARM

SGP observatory using the ArcticShark UAS. The study employed various flight patterns to optimize the integration of ground-based and UAS-borne instruments, focusing on vertically resolved aerosol properties in the SGP region. By combining ARM's UAS capabilities with the established ground-based remote sensing data, this research provides a unique dataset that enables the scientific community to explore atmospheric vertical structures in unprecedented detail. Additionally, insights into aerosol chemical properties at higher altitudes can be obtained through innovative analyses of particle samples collected during UAS

deployments. Overall, with its ability to conduct long-duration flights and carry multiple payloads, the ArcticShark successfully bridged observational gaps and showed great potential to enhance our understanding of vertical atmospheric structures. This integration of UAS and ground-based measurements represents a significant advancement in atmospheric data collection, particularly for studying aerosols and their impacts on weather and climate.

## 2    Data and Measurements

### 2.1    ArcticShark in situ measurements

     The ArcticShark is an advanced mid-size UAS supported by the DOE ARM program to conduct atmospheric research (https://www.arm.gov/guidance/campaign-guidelines/arcticshark). The ArcticShark can carry a scientific payload of up to 45 kg (~100 lbs), which can include a variety of meteorological, aerosol, trace-gas and cloud instruments. The ArcticShark can reach altitudes of up to 5,500 m and has a flight duration of up to 8 hours. This operation range enables data collection over a

large spatial area and extended periods, providing a detailed picture of the atmospheric state. The ArcticShark was intensively operated by the AAF in March, June, and August of 2023, allowing for comprehensive data collection above the ARM SGP observatory and contributing valuable data to the scientific community. Throughout three deployments, the AAF engineering and science flights primarily aimed to comprehend the flight operation envelope and determine the optimal operational parameters. Additionally, these flights carried out the scientific measurements of thermodynamic, aerosol, and land-surface

properties and the exploration of various flight patterns to effectively address various scientific questions.

     The ArcticShark has an interior payload bay of around 85 Liters and four underwing-mounted pylons to carry these various instrument packages. It provides 2500 W of electrical power specifically for operating the scientific payloads, enabling the integration of multiple sensors simultaneously. The typical measurements include atmospheric state and thermodynamic properties (temperature, humidity, pressure, and 3-D wind components), aerosol (total number concentration, size distribution,

optical properties and chemical compositions) and cloud measurements, atmospheric gases (water vapor and carbon dioxide concentrations), and land surface monitoring (infrared surface temperature and multispectral images) (Mei et al., 2022; Mei et



al., 2024) (detailed in Table 1). Although the typical measurements acquire data at a 1 Hz sampling rate, the ArcticShark is also equipped with the advanced meteorological instrument, the Airborne Inertial Measurement and Meteorological System (AIMMS-30), to provide high-frequency measurements. The AIMMS-30 was tested and calibrated under specific maneuver

flight patterns to ensure the accuracy and reliability of the data collected during the first flight of each mission. With the calibration flight and appropriate post-processing, ArcticShark can provide wind data at a rate of 100 Hz to the scientific community (DOI: 10.5439/2204047), which can be used to derive further turbulence parameters, such as turbulence kinetic energy (TKE). Before and after each deployment, the aerosol instruments were calibrated in the lab to ensure counting efficiency and sizing accuracy. During deployment, their performance was checked against AAF standard instruments to

maintain data consistency and high-quality results (Mei et al., 2022).

**Table 1. DOI information of ArcticShark and VAPs datasets**

| ARM data product | Description | DOI |
|---|---|---|
| aafh2o (Burk et al., 2023a) | Airborne measurements of H2O concentrations | https://doi.org/10.5439/1821160 |
| aafirt (Burk et al., 2023b) | Infrared Thermometer (IRT) on airborne platform | https://doi.org/10.5439/1821129 |
| aafnav (Mei et al., 2023a) | ARM Aerial Facility (AAF) Navigation (NAV) Datastream | https://doi.org/10.5439/1339718 |
| aafnavvec (Mei et al., 2023b) | ARM Aerial Facility (AAF) VectorNav, VN-200, GPS-Aided Inertial Navigation System | https://doi.org/10.5439/1238153 |
| aafmcpc (Burk et al., 2023c) | Unmanned Aircraft Systems, Mixing Condensation Particle Counter | https://doi.org/10.5439/1820906 |
| aafpops (Mei et al., 2023c) | Portable Optical Particle Counter | https://doi.org/10.5439/2322345 |
| aafstap (Gibler et al., 2023a) | Single Channel Tricolor Absorption Photometer | https://doi.org/10.5439/1838697 |
| aafmetaims100hz (Mei et al., 2023d) | Integrated Meteorological Measurement System (AIMMS) - 100 Hz Meteorological data | https://doi.org/10.5439/2204047 |
| aaffiltsamp (Burk et al., 2023d) | Unmanned Aircraft Systems, Filter Sampler | https://doi.org/10.5439/1821176 |
| aafmopc (Gibler et al., 2023b) | Miniaturized Optical Particle Counter | https://doi.org/10.5439/1838698 |
| aafnavaims (Cristina et al., 2023) | Integrated Meteorological Measurement System (AIMMS) - Navigation data | https://doi.org/10.5439/1238157 |



| aafmetaims (Koontz et al., 2023) | Integrated Meteorological Measurement System (AIMMS) - Meteorological data | https://doi.org/10.5439/1349241 |
|---|---|---|
| aaftrh (Burk et al., 2023e) | Temperature and Relative Humidity | https://doi.org/10.5439/1820905 |
| aafcdp (Gibler et al., 2023c) | Cloud Droplet Probe | https://doi.org/10.5439/1561461 |
| aafnavaims100hz (Mei et al., 2023e) | Integrated Meteorological Measurement System (AIMMS) - 100 Hz Navigation data | https://doi.org/10.5439/2204048 |
| cldtype (Zhang et. al., 2023) | Cloud Type Classification | https://doi.org/10.5439/1349884 |
| mplcmaskml (Cromwell et. al., 2023) | Micropulse Lidar cloud mask using machine learning model from Cromwell et. al 2019 | https://doi.org/10.5439/1637940 |
| ceilpblht (Morris et. al., 2023 ) | Ceilometer (CEIL): planetary boundary-layer heights | https://doi.org/10.5439/1095593 |
| pblhtrl1zhang (Zhang et. al., 2023) | Planetary Boundary Layer derived from Raman Lidar data using Damao Zhang algorithm | https://doi.org/10.5439/2282350 |
| rlprof-fex (Cromwell et. al., 2023) | Raman Lidar: Aerosol backscatter, scattering ratio, lidar ratio, extinction, cloud mask, and linear depolarization ratio derived from Thorson FEX code | https://doi.org/10.5439/1373934 |
| rnccn (Sivaraman et. al., 2023) | Retrieved Number concentration of CCN profile from Kulkarni 1st algorithm | https://doi.org/10.5439/1813858 |

## 2.2    Offline chemical analysis

The primary advantage of offline chemical analysis is the ability to employ sophisticated laboratory-based analytical techniques impractical for airborne deployment due to payload weight and capacity constraints. The filter samples collected by ArcticShark leverage the advanced chemical analysis capabilities of facilities such as the Environmental Molecular Sciences Laboratory (EMSL), another DOE user facility operated by the Pacific Northwest National Laboratory.

The advanced chemical analysis allows for more comprehensive and detailed analysis of chemical composition to provide deeper insights into the chemical properties of atmospheric particles, including the use of highly sophisticated analytical instruments like a Micro-Nebulization Aerosol Mass Spectrometer (MN-AMS), Computer Controlled Scanning Electron Microscopy with Energy Dispersive X-ray Spectroscopy (CCSEM-EDX), Orbitrap high-resolution mass spectrometry (HRMS), and Time-of-Flight Secondary Ion Mass Spectrometer (TOF-SIMS). The MN-AMS enables highly sensitive quantification of aerosol composition from the UAS-collected filter samples, with detection limits down to nanogram



levels for species like sulfate, nitrate, and organics (Niedek et al., 2022). Combining the MN-AMS technique with other offline

methods like TOF-SIMS provides comprehensive insights into organic aerosol composition, oxidation state, mixing state with inorganics, and source differentiation (e.g., biomass burning vs. biogenic).

Integrating the STAC (Size and Time resolved Aerosol Collector) impactor (Cheng et al., 2022) with the ArcticShark sampler, aerosol samples can also be collected on TEM grids and Silicon nitride (SiNx) substrates. These substrates can be further analyzed using CCSEM-EDX to determine individual particle characteristics, such as size, morphology, mixing state,

water uptake potential and elemental composition. (Cheng et al., 2023) This method offers valuable information about various atmospheric particle types and their potential sources. (Lata et al., 2023) Alternatively, these substrates can be directly analyzed with HRMS coupled with a nanospray desorption electrospray Ionization (nano-DESI) source to elucidate intact organic molecular formulas. Researchers can derive key parameters from the mass spectrometer data, including O:C ratios, carbon oxidation states, aromaticity indices, and organic aerosol volatility distributions. (Roach et al., 2010; Vandergrift et al., 2024;

Vandergrift et al., 2022).

## 2.3    ARM value-added products

To facilitate the use of ARM data more effectively, ARM has developed higher-order data products known as Value-added products (VAPs) (https://www.arm.gov/capabilities/science-data-products/vaps). These VAPs are generated by applying advanced, well-developed retrieval algorithms or implementing additional quality control to existing ARM

datastreams, enhancing the user's scientific research and model development. Over a hundred baseline VAPs currently cover a wide range of atmospheric parameters, including aerosol and cloud macro- and microphysical properties, chemical properties, precipitating retrievals, atmospheric environment and radiation budget, and various modeling VAPs.

In this study, we utilized the CLDTYPE (Flynn et al., 2017) (https://www.arm.gov/capabilities/science-data-products/vaps/cldtype) and the MPLCMASKML (Flynn et al., 2023) (https://www.arm.gov/capabilities/science-data-

products/vaps/mplcmaskml) VAPs for tracking clouds and determining cloud boundaries. For boundary layer height estimations, we overlayed our flight tracks with the CEILPBLHT (Sivaraman et al., 2013) (https://www.arm.gov/capabilities/instruments/ceil) and PBLHTRL1ZHANG (Zhang et al., 2022) (https://www.arm.gov/capabilities/science-data-products/vaps/pblht) VAPs. Additionally, the Raman Lidar RLPROF-FEX (Chand et al., 2022) (https://www.arm.gov/capabilities/science-data-products/vaps/rlprof-fex) VAP was used to obtain aerosol

particulate backscatter coefficients and aerosol extinction coefficients.

The ARMTRAJ-AAF VAP, offering a Lagrangian back-trajectory dataset was also used in this study. This dataset provided detailed information about the coordinates and thermodynamic properties of airmasses prior to their transport to the UAS sampling region. Trajectories are calculated using the Hybrid Single-Particle Lagrangian Integrated Trajectory (HYSPLIT) model informed by the European Centre for Medium-Range Weather Forecasts ERA5 reanalysis dataset at its

highest spatial resolution (0.25 degrees), and are initialized using ArcticShark sampling times and coordinates (latitude,



longitude, and altitude range). Similar to other ARMTRAJ datasets, the ARMTRAJ-AAF provides ensemble run statistics, which are used here as they enhance the trajectory robustness. (Silber et al., 2024).

The Retrieved Number Concentration of CCN VAP (RNCCN, https://www.arm.gov/capabilities/science-data-products/vaps/rnccn) provides hourly vertical profiles of CCN concentration at various supersaturation values (Kulkarni et al., 2023b). The VAP algorithm is based on the Ghan and Collins (Ghan and Collins, 2004) and Ghan et al. (Ghan et al., 2006) methods that scale the surface CCN concentration with the dry extinction profiles. The dry extinction profiles are calculated after removing the influence of humidification from the extinction profiles, and to retrieve the vertical CCN concentration, the VAP assumes that aerosol composition is uniform vertically and larger aerosol (> 100 nm) induces droplet activation first.

## 3 Results

### 3.1 Overview of the airborne observations

#### 3.1.1 Flight tracks

The AAF deployments at the SGP site consisted of a series of flights designed to gather data on the optimal operational parameters under various atmospheric conditions. The diverse flight tracks ensured comprehensive scientific data collection across different geographical areas and weather systems. Figure 1 illustrates the flight tracks from 2021 to 2023, highlighting an extension of the sampling areas in August 2023. This August expansion (flight track in white color) is notably larger compared to the flights conducted before August (represented in light blue), which relied on ground-based visual observers (VO). This improvement in flight range is attributed to operational advancements enabled by having a VO aboard the chase plane. Previously, the UAS was restricted to the red Certificate of Authorization (COA) area with the ground VO. With permission to reach into the yellow COA area, ArcticShark operated in a larger area and reached high altitudes in the dark blue area where the UAS can reach up to 5,500 meters. It allows the UAS to gather data from higher altitudes, which can be crucial for studying the planetary boundary layer and the lower troposphere. It also indicated a robust performance of the UAS in terms of altitude range, as shown in Figure 2.





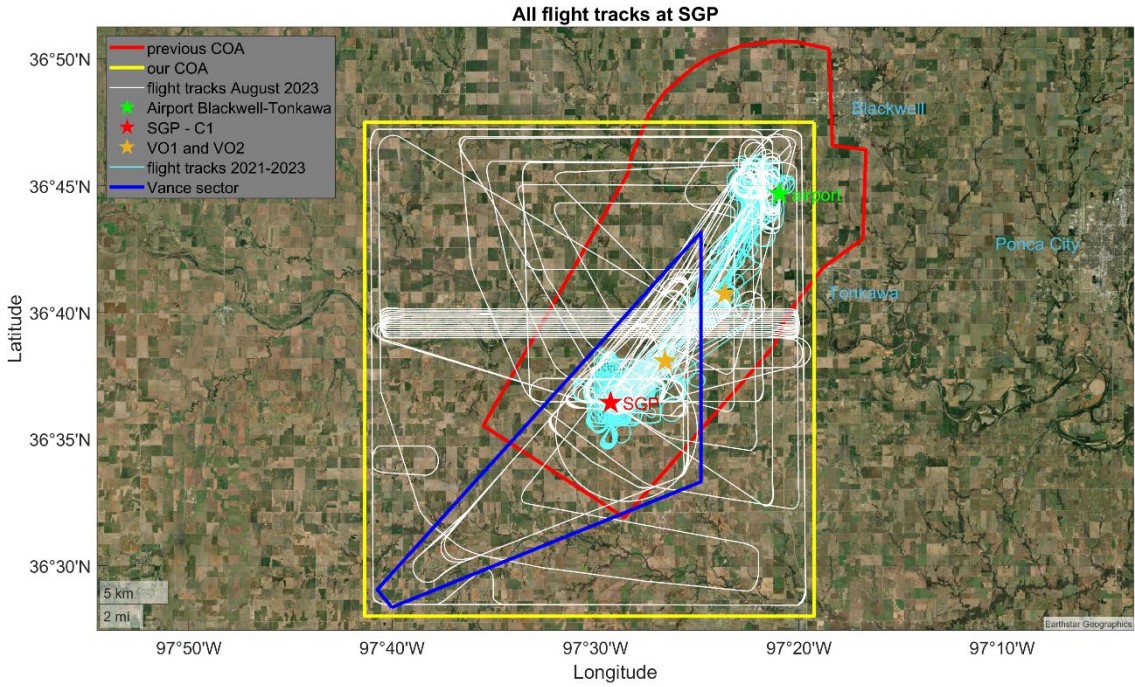

**Figure 1. All ArcticShark flight tracks above the SGP central facility between 2021 and 2023. The COA area expanded from the red box in 2021 to the yellow box in 2023. Due to airspace restrictions, flights above 6000 ft (1828.8 m) are permitted only inside the blue triangle. The white flight tracks show the UAS flight range in August 2023 with the chase plane. The light blue flight tracks show the sampling range between 2021 and 2023 with the visual observers on the ground (green, orange, and red asterisks).**

### 3.1.2 Measurements in March, June, and August 2023

The March, June, and August 2023 flights provided vertical meteorological information from the airborne measurements, as shown in Figure 2 and Figures S1, S2, and S3. In Figure 2, the data was averaged within altitude intervals of 100 meters for March and June flights and 500 meters for August flights. As shown in Fig 2 (a), the ambient temperature decreases as expected with the increase in altitude. The average temperature in March was around 5 degrees Celsius, typical for the tail end of winter and the beginning of spring. By June, the average temperature had increased significantly, reflecting the onset of summer. By August, the average temperature reached nearly 30 degrees Celsius at the lowest flight level, indicating the peak of the summer season. The relative humidity (Fig 2 (b)) showed a similar range across all three months up to 2000 m above the ground but showed more variation in March (Figure S1). This could be due to the transition from winter to spring, which could bring a mix of weather conditions and, therefore, a wider range of humidity levels. Above 2000 meters, relative humidity (RH) values in August increased and exhibited considerable variation, probably due to air cooling, proximity to moisture sources, and atmospheric dynamics. With the chase plane, ArcticShark can fly through holes in broken cloud fields



and reach altitudes above the cloud tops, allowing it to operate in areas with higher moisture content, closer to the air's saturation point.

**Figure 2. Atmospheric conditions encountered during the March, June, and August 2023 flights. (a) ambient temperature; (b)**
**ambient relative humidity; (c) total number concentration from the mixing condensation particle counter (CPC); and (d) total**

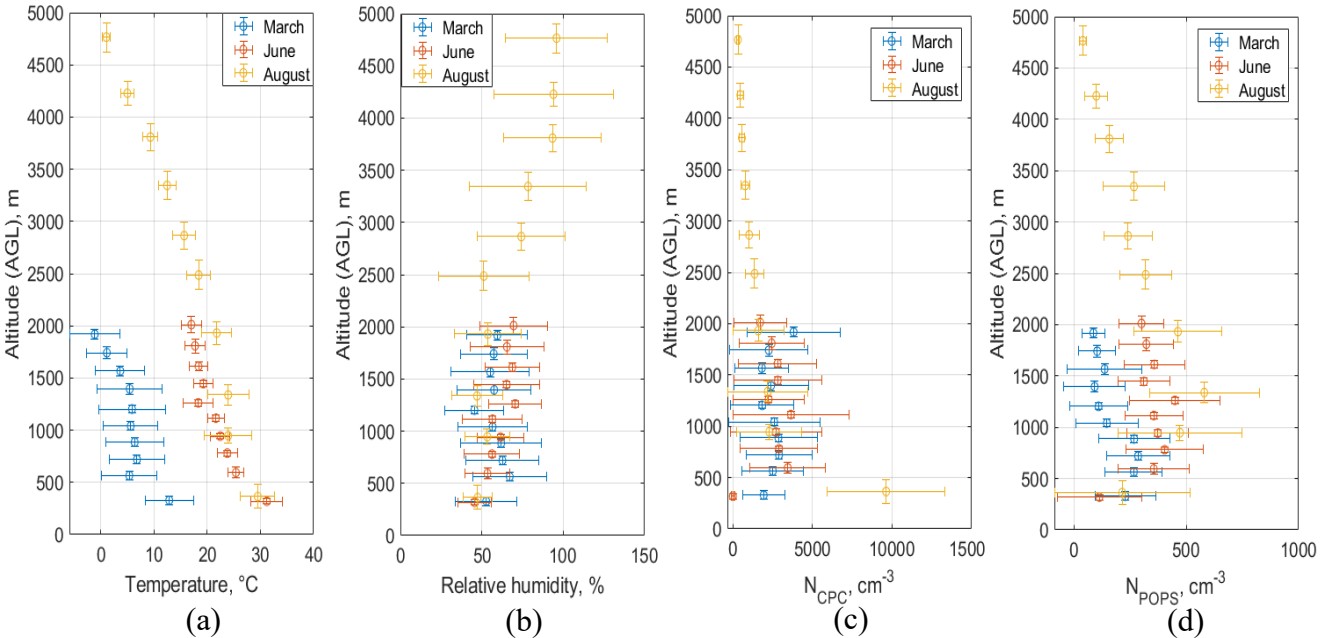

(a)  (b)  (c)  (d)

**number concentration from the portable optical particle spectrometer (POPS).**

The total number concentrations of ambient particles remain relatively stable across all three months within the 500 to 2000 m altitude range and decrease with the increase in altitude, as shown in Fig. 2 (c). This consistency suggests that the
overall particle load in the atmosphere at these elevations does not vary significantly in those three months. Meanwhile, near the surface, we observed a notable increase in particle concentration close to the ground in August, which might be related to the haze environment prevalent during that month and local agricultural burning events. In contrast, the number concentration of larger particles, specifically those with a diameter greater than 135 nanometers (as shown in Fig. 2(d)), rose steadily from March to August. In March, the concentration of these larger particles was relatively lower, which might indicate a slower
growth rate. This slower growth could be linked to the colder temperatures typical of early spring, which may have inhibited atmospheric aerosol particle growth or source activities responsible for forming and accumulating larger particles. As temperatures warmed from March through August, the increase in particle concentration, especially in the accumulation size range, could reflect enhanced atmospheric processes, such as more active secondary particle formation or increased emissions



from local agriculture sources. The warmer temperatures likely facilitated these processes, leading to the observed rise in larger

particles as the months progressed.

Figure 3 compares meteorological data collected by the ArcticShark with the data collected by a weather balloon when both were in the air for March, June and August flights. These two platforms offer different advantages and can provide complementary information about atmospheric conditions. Weather balloons are among the most straightforward and cost-effective tools for atmospheric measurements, with a well-established history of providing consistent long-term data (Vömel

and Ingleby, 2023). In contrast, UAS enables more targeted data collection, allowing for the simultaneous operation of multiple sensors to gather diverse datasets during the same airborne mission. In this study, we used orthogonal linear regression to fit a linear model to the data because there are measurement errors in both ArcticShark and weather balloon measurements. Strong agreement was observed between the ArcticShark and weather balloon data for ambient temperature and humidity, with slopes near 1 and high R-squared values indicating strong correlations, as shown in Figures 3(a) and 3(b). ArcticShark was equipped

with redundant temperature sensors (AIMMS-30 and a fiber-optic thermal sensor) and humidity sensors (AIMMS-30 and a LiCor $H_2O/CO_2$ analyzer), both of which showed strong agreement with the weather balloon measurements.

Additionally, the study found a good correlation ($R^2 > 0.92$) for wind speed and direction comparison between ArcticShark and the weather balloon data. These agreements confirm that theArcticShark's sensors accurately captured the atmospheric conditions at various altitudes, validating its use for meteorological research. While weather balloons remain the

standard for high-altitude measurements, UAS are emerging as a valuable complementary tool, offering flexibility, reusability, and high spatial resolution measurements.



(a)  (b)

(c)  (d)

**Figure 3. Meteorological data comparison between the ArcticShark flight (for March, June, and August) and the weather balloon 2023. (a) Dew temperature comparison, AIMMS-30 and LICOR aboard ArcticShark; (b) Temperature comparison, AIMMS-30 and the Fiber optical thermal sensor aboard ArcticShark; (c) Wind direction comparison; and (d) Wind speed comparison.**

Integrating ArcticShark flight data with ARM remote sensing observations (Figures 4 and 5) offers valuable insights into the varying atmospheric conditions encountered across different months, how these conditions influence UAS operations, and the data types collected. Figure 4 overlayed the cloud masks (from MPLCMASKML VAP), flight altitude, and Planetary Boundary Layer (PBL) height from three typical March, June, and August flight days. The height of the PBL varies with the seasons, generally lower in the spring due to less intense solar heating of the Earth's surface and higher in the summer due to increased solar heating. As shown in Figure 4, the PBL height in March and June was generally lower compared to August



due to several factors: lower solar radiation and surface heating in early spring, the stabilizing temperature gradient, and
different atmospheric dynamics. In March and June, increased moisture and still-growing vegetation contribute to lower
sensible heat flux. Conversely, August typically experiences intense solar heating, stronger convective currents, and drier
conditions, all leading to a higher PBL.

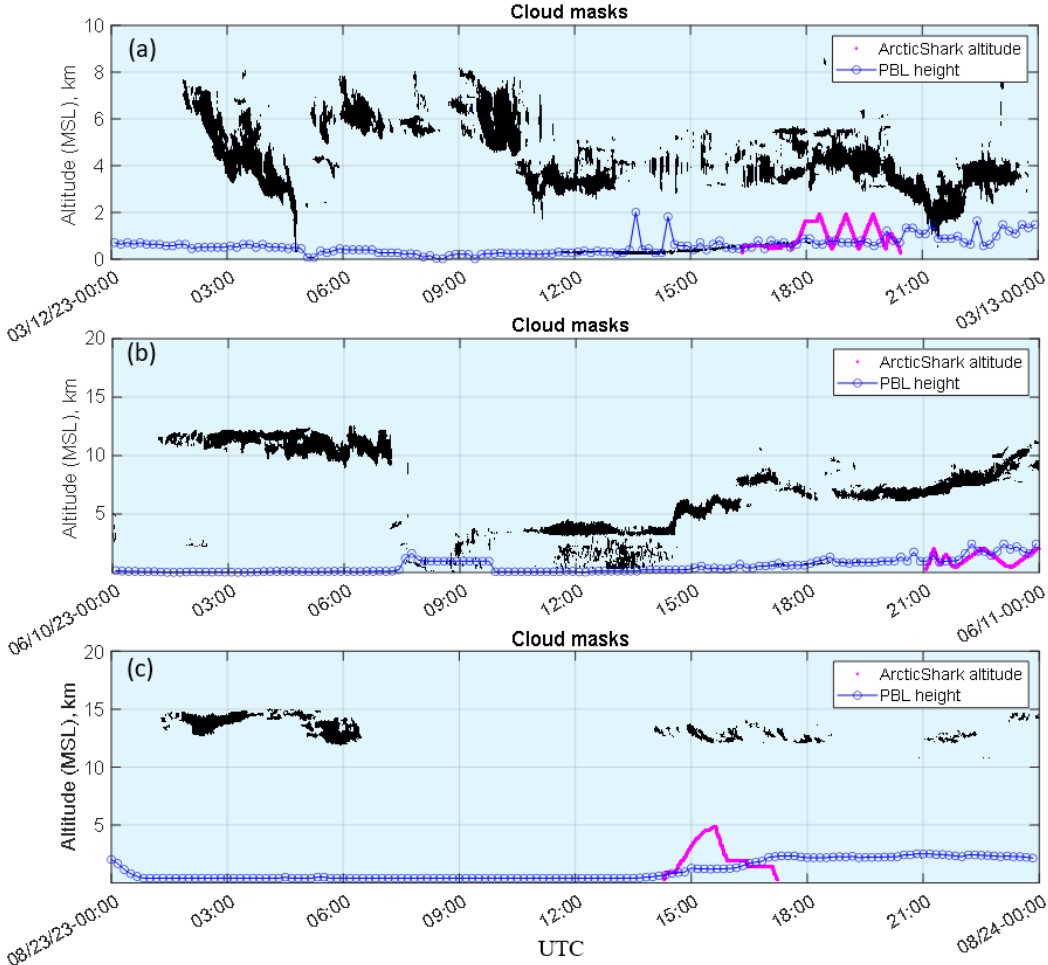

**Figure 4. Typical ArcticShark flight altitude overlay with the cloud masks and the planetary boundary layer height on (a) March**
**12, (b) June 10, and (c) August 23. The y-axis is the altitude above the mean sea level (MSL).**

Additionally, the study found a good correlation ($R^2 >0.92$) for wind speed and direction comparison between
ArcticShark and the weather balloon data. These agreements confirm that theArcticShark's sensors accurately captured the
atmospheric conditions at various altitudes, validating its use for meteorological research. While weather balloons remain the





standard for high-altitude measurements, UAS are emerging as a valuable complementary tool, offering flexibility, reusability, and high spatial resolution measurements.

Integrating ArcticShark flight data with ARM remote sensing observations (Figures 4 and 5) offers valuable insights into the varying atmospheric conditions encountered across different months, how these conditions influence UAS operations,
and the data types collected. Figure 4 overlayed the cloud masks (from MPLCMASKML VAP), flight altitude, and Planetary Boundary Layer (PBL) height from three typical March, June, and August flight days. The height of the PBL varies with the seasons, generally lower in the spring due to less intense solar heating of the Earth's surface and higher in the summer due to increased solar heating. As shown in Figure 4, the PBL height in March and June was generally lower compared to August due to several factors: lower solar radiation and surface heating in early spring, the stabilizing temperature gradient, and
different atmospheric dynamics. In March and June, increased moisture and still-growing vegetation contribute to lower sensible heat flux. Conversely, August typically experiences intense solar heating, stronger convective currents, and drier conditions, all leading to a higher PBL.

Additionally, the figure showed that the clouds on fight days in March were much lower than in June and August, which aligns with the lower PBL height in the spring. In March, cumulus congestus clouds were most common in the region
and more common in spring's cooler, more variable weather. Cumulus and convective clouds are observed for flight days in June. These clouds were typically associated with warm weather and were more prevalent as our flights moved into the summer months. By August, cirrus clouds were the dominant cloud type on flight days. These are high-altitude clouds that form above 6,000 meters and are often associated with fair weather, which is favored for the UAS flight operation.

The combination of the backscattering coefficient (from RLPROF-FEX VAP), flight altitude, PBL height, and TKE
(estimated based on ArcticShark measurement) provides a comprehensive picture of the composition and structure of the atmosphere, as shown in Figure 5. The figure showed that the TKE values were nearly zero when the ArcticShark flew above the PBL. When the ArcticShark was within the PBL, the TKE values significantly increased. As expected, the turbulence intensity should be higher within the PBL because this is where the sun's heating of the Earth's surface generates thermal turbulence. This observation is particularly useful in August when reliable measurements of the PBL height are unavailable.
Vertical gradients in TKE can indirectly indicate the PBL height, as the boundary between the turbulent and non-turbulent regions of the atmosphere corresponds to the top of the PBL. Therefore, by observing where the TKE values increase, we can infer the height of the PBL. The backscattering coefficients exhibited varying ranges from March to August, with values in March being ten times higher compared to those in August. An aerosol layer was aloft at the beginning of the March flight (Figure 5(a)) above the SGP observatory. However, the aircraft was flown between the SGP site and Blackwell airport and did
not capture more information about that layer. The backscattering plot captured the residue layer in the June flight while the ArcticShark flew into it between 23:00 and 23:30 UTC.



**Figure 5. Aerosol backscattering overlayed with the ArcticShark flight altitude, the PBL height and TKE values on (a) March 12, (b) June 10, and (c) August 22. The y-axis is the altitude above mean sea level (MSL).**








**Figure 6. ARMTRAJ-AAF 5-day back trajectory properties on March 12 (a) and (b), June 10 (c) and (d), and August 23 (e) and (f). The illustrated ensemble mean trajectories are calculated based on the flight coordinates and altitude range during measurement periods in (a), (c), and (e) and based on the ground level at the ArcticShark's flight coordinates in (b), (d), and (f). The star is the**

**central facility at the SGP site. All figures are generated using Natural Earth by MATLAB®.**

The back trajectory of airmasses can support aerosol studies by providing context to long-range aerosol transport and suggest potential interactions during their path. Figure 6 presents a comparison of ARMTRAJ 5-day back-trajectory properties on three separate dates: March 12, June 10, and August 23. For each date, two types of trajectories are shown: one set based

on flight coordinates and altitude range during measurement periods (Figures 6(a), 6(c), and 6(e)) and another set of trajectories initialized at the ground level at the ArcticShark's flight coordinates (Figures 6(b), 6(d), and 6(f)). The ensemble statistics presented here are based on 25-member ensembles generated for each trajectory initialization altitude over a 5x5 grid typically spanning several kilometers to each direction relative to the ArcticShark's coordinates. The ensemble mean trajectories calculated using the flight altitudes (Figures 6(a), 6(c), and 6(e)) indicate longer travel pathway distances compared to those

calculated based on ground-level initialization. The trajectories suggest that the flight period measurements included a wider range of atmospheric conditions and altitudes, capturing more variant and extended airmass pathways compared to ground-level measurements that were more localized. On March 12, The airmass trajectories originated mainly from the north region of the US before reaching the sampling area near the Southern Great Plains (SGP) site. The trajectories showed that the airmass traveled from the southwest on June 12. On August 23, the airmass was more influenced by the southeast region. The

differences in airmass origin and trajectory paths between the three dates could be attributed to seasonal atmospheric circulation patterns, which vary with changes in temperature, pressure systems, and overall weather conditions.

**Table 2. Chemical composition of UAS filter samples measured by the MN-AMS analysis**

| Start time | End time | Ambient Mass concentration ($\mu$g/m$^3$) | Volume fraction | | | O/C | H/C | Organic density calculated (kg/m$^3$) | $\kappa_{Org}$ | $\kappa_{Overall}$ |
| | | | (NH$_4$)$_2$SO$_4$ | NH$_4$NO$_3$ | Organics | | | | | |
|---|---|---|---|---|---|---|---|---|---|---|
| 3/9/23 21:13 | 3/9/23 23:04 | 6.0 | 0.06 | 0.19 | 0.75 | 0.3459 | 1.6817 | 1141 | 0.10 | 0.24 |
| 3/10/23 16:43 | 3/10/23 21:15 | 5.3 | 0.10 | 0.32 | 0.58 | 0.4758 | 1.6038 | 1249 | 0.20 | 0.39 |
| 3/12/23 16:20 | 3/12/23 20:27 | 3.2 | 0.06 | 0.26 | 0.68 | 0.3358 | 1.6897 | 1132 | 0.09 | 0.27 |
| 3/13/23 15:26 | 3/13/23 18:01 | 3.5 | 0.07 | 0.13 | 0.80 | 0.3061 | 1.7107 | 1106 | 0.06 | 0.18 |



| 3/14/23 17:25 | 3/14/23 21:28 | 3.2 | 0.08 | 0.19 | 0.74 | 0.3555 | 1.6626 | 1153 | 0.11 | 0.25 |
|---|---|---|---|---|---|---|---|---|---|---|
| 3/17/23 21:30 | 3/17/23 23:39 | 4.3 | 0.08 | 0.03 | 0.89 | 0.3074 | 1.7134 | 1106 | 0.07 | 0.13 |
| 6/8/23 16:28 | 6/8/23 19:07 | 5.2 | 0.10 | 0.02 | 0.88 | 0.4669 | 1.6102 | 1241 | 0.20 | 0.25 |
| 6/9/23 13:49 | 6/9/23 17:40 | 5.9 | 0.08 | 0.03 | 0.90 | 0.4310 | 1.6497 | 1206 | 0.17 | 0.22 |
| 6/10/23 21:00 | 6/11/23 1:06 | 27 | 0.05 | 0.01 | 0.93 | 0.5367 | 1.6438 | 1274 | 0.26 | 0.28 |
| 6/12/23 17:45 | 6/12/23 19:42 | 6.9 | 0.05 | 0.02 | 0.93 | 0.4019 | 1.6849 | 1177 | 0.14 | 0.18 |
| 6/22/23 14:13 | 6/22/23 16:58 | 8.5 | 0.07 | 0.02 | 0.91 | 0.4624 | 1.6463 | 1227 | 0.19 | 0.23 |
| 6/23/23 14:32 | 6/23/23 16:13 | 8.7 | 0.09 | 0.03 | 0.89 | 0.4279 | 1.6697 | 1198 | 0.17 | 0.22 |
| 8/17/23 14:58 | 8/17/23 17:40 | 8.8 | 0.20 | 0.07 | 0.73 | 0.3316 | 1.8754 | 1080 | 0.09 | 0.23 |
| 8/18/23 14:04 | 8/18/23 17:44 | 7.6 | 0.10 | 0.04 | 0.87 | 0.3330 | 1.8764 | 1081 | 0.09 | 0.16 |
| 8/21/23 14:45 | 8/21/23 17:08 | 14 | 0.12 | 0.04 | 0.84 | 0.2991 | 1.9034 | 1052 | 0.06 | 0.15 |
| 8/22/23 14:45 | 8/22/23 17:56 | 8.9 | 0.09 | 0.03 | 0.87 | 0.3217 | 1.8876 | 1071 | 0.08 | 0.15 |
| 8/23/23 14:14 | 8/23/23 17:39 | 14 | 0.07 | 0.03 | 0.90 | 0.3109 | 1.9047 | 1060 | 0.07 | 0.12 |
| 8/24/23 13:40 | 8/24/23 17:43 | 7.5 | 0.09 | 0.03 | 0.88 | 0.3064 | 1.9069 | 1056 | 0.06 | 0.13 |
| 8/26/23 13:51 | 8/26/23 19:38 | 10 | 0.08 | 0.03 | 0.89 | 0.2910 | 1.9073 | 1046 | 0.05 | 0.11 |
| 8/27/23 13:58 | 8/27/23 17:14 | 8.8 | 0.12 | 0.04 | 0.83 | 0.3100 | 1.8935 | 1062 | 0.07 | 0.16 |



| | | | | | | | | | | |
|---|---|---|---|---|---|---|---|---|---|---|
| 8/29/23 15:18 | 8/29/23 19:49 | 39 | 0.13 | 0.05 | 0.82 | 0.4321 | 1.8664 | 1146 | 0.17 | 0.25 |
| 8/30/23 15:17 | 8/30/23 20:32 | 28 | 0.29 | 0.11 | 0.60 | 0.4766 | 1.7565 | 1204 | 0.21 | 0.37 |

## 3.2    Case study with unique measurement capabilities

### 3.2.1    Advanced offline chemical analysis

Based on the chemical analysis results of the MN-AMS, we summarized the chemical compositions derived from the flights in Table 2, which displayed a clear seasonal trend in the chemical composition data from the ARM SGP site. The total mass loading increased from March to August, consistent with the trend in aerosol total number concentrations. In March, the average total mass concentration of organic species and ammonium salts was 4.2 µg/m$^3$, doubling to 10.4 µg/m$^3$ in June and escalating to 14.7 µg/m$^3$ in August. This pattern is consistent with seasonal differences observed in previous studies (Fast et al., 2022; Liu et al., 2021; Parworth et al., 2015). Interestingly, the organic volume fraction in the samples from March was lower (typically less than 80%) than those from June. The June samples exhibited a higher oxygen-to-carbon (O:C) ratio, signifying that the organic aerosols were more oxidized. This trend can be attributed to the increased atmospheric oxidative reactions during warmer weather. The presence of more organic aerosols in the atmosphere in June and August could also result from increased biological activity during the summer months. This diverse chemical composition explains the increased variability (O:C ratios and organic volumetric fractions) observed in the August samples.

The integration of samples collected from the ArcticShark with advanced offline high-resolution analytical techniques is shown in Figure 7 for the flight on June 19, 2023 (continuous collection from 600 – 2000 m above sea level). For this representative sample, analysis via the nano-DESI HRMS pipeline resulted in 767 individual molecular formulas (MF) assignments, including a high proportion of organosulfates (99 MF containing C, H, O, and S atoms) and organonitrates (230 MF containing C, H, N, and O atoms; mass spectrum shown in Figure 7(a)). The assigned MF are then parametrized according to the strategy from Li et al. (Li et al., 2016), resulting in a volatility distribution (individual MFs are classified as volatile organic carbon (VOC), intermediate VOC (IVOC), semi-VOC (SVOC), low VOC (LVOC), or extremely low VOC (ELVOC); Figure 7(b)). For the same sample, Figure 7 (c) shows an exemplary top-view scanning electron microscopy (SEM) image, showing the dominance of organic particles and the potential for K$_2$SO$_4$ inclusion within the organic particles. Figure 7 (d) depicts the size-resolved chemical composition (acquired via CCSEM/EDX indicating dominance of carbonaceous (CNO, 38.4%) and carbonaceous sulfate (CNOS, 61.1%) aerosol with minor fraction of K$_2$SO$_4$ (0.4%) containing aerosol. The particle classification scheme was illustrated in Figure S4. More studies on the chemical characterization of the 2023 flight samples or size-resolved compositions were under preparation. (Niedek; Mansoura)





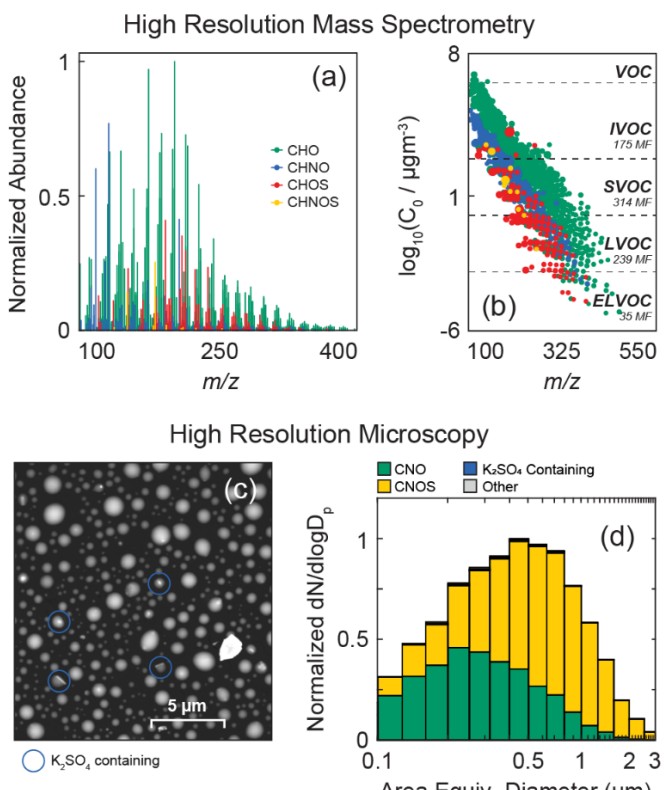

335

**Figure 7. Offline high-resolution analyses of the molecular composition of organic aerosols and size-resolved chemical composition for a representative sample from the June 19, 2023 flight (continuous sample collection over 600 – 2000 m MSL) (a) mass spectrum from direct nano-DESI HRMS analysis; (b) volatility distribution from parametrized mass spectrum data; (c) size-resolved chemical composition from CCSEM/EDX; (d) top-view SEM image, highlighting instances of inorganic inclusion.**

340    **3.2.2    Vertical profile of aerosol optical properties**

With an overview of the atmospheric parameters, the following sections explore two case studies that further illuminate our UAS capabilities – providing detailed vertical information on aerosol optical properties and aerosol's potential to form clouds (Cloud Condensation Nuclei (CCN) concentrations), which enable a more accurate and comprehensive assessment of aerosol impacts on the Earth's radiation budget.





**Figure 8. Aerosol extinction coefficient comparison with the Raman Lidar (RL) retrieval on Aug. 22, 2023. Note that the altitude is above ground level. The estimated aerosol extinction coefficients (a) were under three conditions: the sampling dry condition, corrected with the averaged ambient RH condition, and corrected with the ambient RH profile condition (b).**

Aerosol optical properties depend on relative humidity, aerosol size distributions (usually measured at RHs lower than ambient RH), and the complex refractive index, which should be adjusted accordingly. (Ghan and Schwartz, 2007; McComiskey and Ferrare, 2016) In this section, we discuss our approaches to estimating aerosol optical profiles under ambient conditions, which involve accounting for various factors (e.g., ambient temperature, pressure, and RH) that influence how aerosol interacts with light. The aerosol profiles of the extinction coefficients are shown in Figure 8(a). The ambient RH profile is shown in Figure 8(b). All the extinction coefficient values were derived under the ambient temperature and pressure. That allows us to focus on the ambient RH effect on this aerosol optical property. Changes in RH can significantly alter aerosol



size, chemical composition, and refractive index. In this study, we assume that the RH effect on the refractive index and composition is negligible and only consider the effect on the size distribution.

During the airborne sampling, aerosol particles were dried lower than 40% in the inlet manifold. Using the size distribution directly from the portable optical particle spectrometer (POPS), we can derive the dry aerosol extinction coefficient, as shown in Figure 8(a) (blue symbol). The result is consistent with the previous study under low RH (<40%) – the aerosol extinction decreased with altitude increase. (Andrews et al., 2011; Andrews et al., 2004) Then, two approaches were used to study the influence of the RH on the estimated aerosol optical profiles. The first one used the averaged RH value of the profile (based on the right panel of Figure 8(a)), and the growth factor (GF) was calculated as a function of this averaged RH ($RH_{avg}$) value and hygroscopicity (equation 11 in Petters and Kreidenweis's paper, 2008, and Kappa from Table 2), which used the chemical analysis results from the MN-AMS. Then, we assumed that the same GF would weigh the whole size distribution and used the weighted size distribution to estimate the extinction under the ambient RH condition (light brown symbol). The second approach used the f(RH) profile correction (RH profile with magenta symbol). This correction was performed by applying the f(RH) parameterization (Zieger et al., 2011) to the estimated aerosol extinction profile based on the POPS size distribution. (Mei et al., 2024) The fitted gamma parameter ($\gamma = 1.53$, as shown in Figure S5) in this parameterization was obtained from the bulk dataset of all collocated extinction and RH profiles in time with the aircraft sampling periods during the June deployment. The black symbol depicts the retrieved values from Raman Lidar (RLPROF-FEX) at 355 nm wavelength. The comparison showed a good agreement between the aerosol extinction profiles corrected for relative humidity (RH) and the extinction profiles retrieved from lidar in Figure 8. This agreement emphasizes the significant impact of ambient RH on the aerosol extinction coefficients.

This study highlights several promising avenues for future research. Firstly, leveraging UAS to estimate aerosol optical profiles and validate lidar retrievals presents a valuable opportunity. UAS can provide high-resolution vertical profiles and targeted measurements in specific areas of interest, complementing the broader spatial coverage of lidar systems. Additionally, integrating high-resolution sensors for relative humidity (RH) and temperature on UAS platforms represents a significant advancement in vertical atmospheric profiling. Furthermore, combining UAS-borne measurements with lidar retrievals can greatly enhance aerosol research. While lidar systems offer continuous data, UAS provides detailed snapshots at various altitudes, contributing to improved temporal resolution. This synergistic approach not only refines the accuracy of aerosol optical profiles but also introduces a versatile and comprehensive methodology for atmospheric studies.

### 3.2.3    Vertical profile of CCN concentration (CCNc)

Understanding the vertical distribution of CCNc is essential for elucidating how aerosols influence cloud formation and properties throughout the atmospheric column. CCN is pivotal in the nucleation process, affecting both the formation and characteristics of cloud droplets. Specifically, higher CCN concentrations result in numerous smaller droplets, whereas lower concentrations lead to fewer, larger droplets. These variations in droplet size significantly impact cloud albedo, cloud lifetime, and precipitation processes (Li et al., 2022; Seinfeld et al., 2016; Rosenfeld et al., 2014). Detailed CCNc profiles, particularly



at the cloud base where air is predominantly entrained into the cloud, are crucial for accurately assessing aerosol-cloud
interactions (ACI) (Bellouin et al., 2020). We can evaluate and refine model predictions by examining these profiles, especially
at the cloud base. Discrepancies between observed and predicted CCN concentrations can reveal areas where models may need
adjustments and lead to ultimately improving the accuracy of CCN predictions and their integration into climate models.

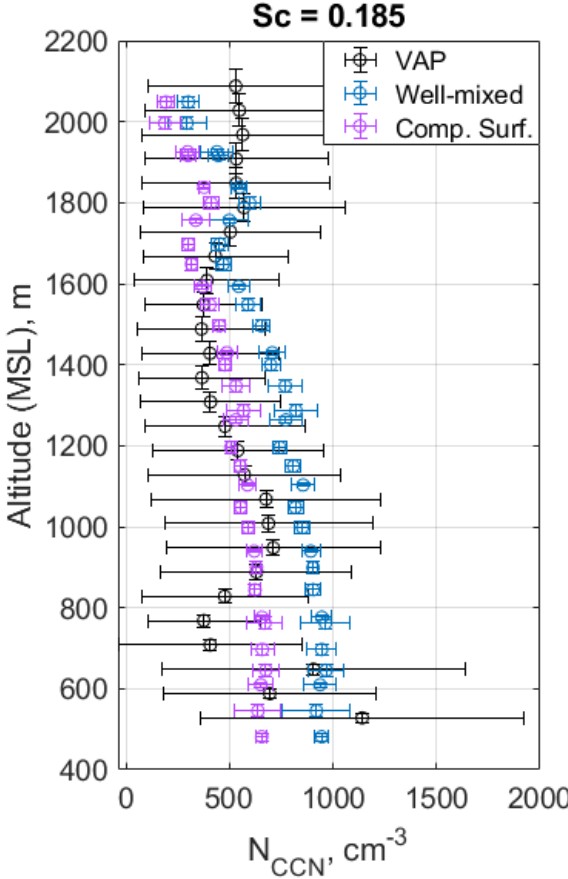

**Figure 9. Cloud condensation nuclei (CCN) concentration ($N_{CCN}$) comparison with the CCNc profile from RNCCN VAP on Aug. 22,**
**2023. The estimated CCN concentrations were under two conditions: 1) chemical species in the CCN population were well-mixed**
**(Well-mixed), and 2) the surface activity can be explained by a compressed-film model (Comp. Suf.).**

This study estimated the CCNc profile based on the in situ aerosol size distribution data from a portable optical particle
spectrometer (POPS) and the chemical composition derived from offline MN-AMS analysis. As shown in Figure 9, two
approaches were used to derive the CCNc profiles. The first approach assumed that the aerosol particles were well-mixed. This
assumption is based on the premise that the aerosol particles are homogeneously distributed within the air mass, and their
chemical and physical properties are uniform throughout the measured size range. This approach allows for estimating the
CCNc profile using the $\kappa$-Kohler theory (Mei et al., 2013; Petters and Kreidenweis, 2007; Thalman et al., 2017; Kulkarni et
al., 2023a). The second approach assumes the formation of a compressed film in the growing droplet, leading to surface tension




depression by interfacial organic molecules (Lowe et al., 2019; Ruehl et al., 2016). The second approach considers the potential
influence of organic compounds on CCN activity. Organic molecules in the aerosol particles can migrate to the particle-water interface, forming a compressed film that can significantly reduce the droplet's surface tension, thereby enhancing the droplet size observed at activation. The well-mixed assumption led to 30-50% more CCN concentration prediction than the compressed surface assumptions. While overlaying the ARM RNCCN VAP data with two profiles, we noticed that both are within the uncertainty range of the CCNc profiles derived by the ARM RNCCN VAP.

Current estimates of CCNc profiles are constrained by the size range of the POPS, which only measures particles larger than 135 nm in diameter. This limitation restricts the ability to estimate the CCN concentrations at the higher supersaturation range (often limited to less than 0.2% for most flight conditions). This narrow supersaturation range can lead to inaccuracies in estimating CCN concentrations, particularly for smaller particles that may play a significant role in cloud nucleation. To address this limitation and provide more accurate CCN profile estimations, AAF has incorporated a miniaturized scanning
electrical mobility spectrometer (mSEMS) to extend the measurement range to include smaller aerosol particles (approximately 10 nm). The mSEMS, POPS, and a custom-built water CPC (with particle collection capability for chemical composition analysis) form an additional payload package designed to study aerosol size distribution and its applications in atmospheric research. Note that the accuracy of estimated CCN profiles is often uncertain due to the reliance on indirect measurements and assumptions. To quantify this uncertainty and assess the limitations of current estimation methods, it is also desirable to
compare estimated CCN profiles with direct in situ CCN measurements with a piloted aircraft campaign. A comparison study can thoroughly evaluate the estimation accuracy, identify discrepancies between estimated and observed CCN concentrations and highlight potential sources of error in the estimation methods.

## 4    Conclusions

     This study summarizes measurements obtained during the ArcticShark deployments to the SGP observatory in March,
June, and August 2023. We provided an overview of the typical atmospheric conditions observed across these seasons, including temperature, relative humidity, aerosol particle concentrations, and chemical compositions. The data reveal significant seasonal variations: temperature and total mass loading increased from March to August, with a notable rise in oxidized and hygroscopic organic aerosols observed in June. Notably, there was strong agreement between temperature and humidity recorded by sensors from the weather balloon and ArcticShark, which indicated the high correlations among various
sensors for critical meteorological parameters, including temperature, humidity, wind speed, and wind direction.

     Analysis of cloud masks, flight altitudes, and planetary boundary layer (PBL) heights for representative flights from each month illustrated that clouds in March were considerably lower in altitude than those in June and August, with distinct cloud types observed in each period. In contrast, higher turbulence was observed within the PBL, as indicated by increased turbulent kinetic energy (TKE) values in June.



435       Understanding these trajectory differences based on ARMTRAJ-AAF helps interpret the aerosol, cloud, and meteorological measurements recorded by ARM facilities. Recognizing the airmass origins provides insight into potential sources of aerosols or precipitation patterns, impacting cloud formation mechanisms and radiative properties.

       We analyzed several cases to further demonstrate the ArcticShark's measurement capabilities. On June 19, 2023, advanced offline analyses of field-collected particles using high-resolution microscopy and mass spectrometry provided
detailed insights into particle size, morphology, and composition. Based on the August 22, 2023 data, we compared aerosol extinction coefficient profiles obtained from lidar with those estimated from airborne measurements, as shown in Figure 8. The optical comparison indicated good agreement between the lidar-retrieved extinction profiles and those corrected for relative humidity. Similarly, Figure 9 demonstrates that CCNc profiles derived from airborne measurements closely matched the ARM RNCCN VAP data, highlighting the potential of using airborne data to validate the remote sensing retrieval
techniques through further in-depth studies.

       In summary, the ArcticShark has proven its capability to collect vertically resolved data under the diverse atmospheric conditions at the ARM SGP site. Integrating UAS data with ground-based observations has provided critical datasets to study atmospheric parameters, aerosol concentrations, chemical composition, and turbulence within the boundary layers. Future work will focus on leveraging both ground-based and airborne measurements, as well as remote sensing techniques, to advance
atmospheric research.

**Acknowledgments**

This research was supported by the ARM user facility, a US DOE Office of Science user facility managed by the Biological and Environmental Research (BER) program. A portion of this research was performed on project awards (10.46936/lser.proj.2020.51377/60000185 and 10.46936/expl.proj.2021.60186/60008210) under the user projects at the Environmental
Molecular Sciences Laboratory (EMSL). Battelle operates the Pacific Northwest National Laboratory (PNNL) for the DOE under contract DE-AC05-76RL01830 to support both EMSL and ARM user facilities. JF acknowledges support from ASR under the Integrated Cloud, Land-surface, and Aerosol System Study (ICLASS) Science Focus Area. QZ and CRN also acknowledge DOE funding from DE-SC0022140. CRN also acknowledges the DOE Office of Science Graduate Student Research (SCGSR) award to support his research and collaboration between UC Davis and PNNL. We acknowledge the use of OpenAI's ChatGPT to assist in refining the language used in this
document.

**Data Availability:**

Data described in this manuscript can be accessed at https://adc.arm.gov/essd/ACP_Mei under each data DOI listed in Table 1. Data collected from the various observation platforms in the SGP region are made freely accessible to the scientific community through public repositories (https://adc.arm.gov/discovery/#/results), facilitating a wide range of research
endeavors.



**Author Contribution:**

FM led the formulation of this paper. FM, GK, and GWV provided the figures. QZ, FM, MSP, JMT, FM, MSP, CRN, SG, BS, and HSM participated the field study and collected the data. XM ZC, GWV, NNL, WC, and ZZ analyzed the chemical samples. FM wrote the draft and all coauthors provided editing suggestions to the manuscript.

**Competing Interests.**

The authors declare that they have no conflict of interest.

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
