# Peer review of "Measurement Report: Vertically resolved Atmospheric Properties Observed over the Southern Great Plains with Uncrewed Aerial System - ArcticShark"

_EGUsphere, 2024_

## Referee Comment (RC2)

[revised manuscript text omitted]

*too low resolution*

*colors are difficult*

*colors?*

*MSL? rename in legend*

*(right y-axis)*

*2023* *left*

**Figure 5. Aerosol backscattering overlayed with the ArcticShark flight altitude, the PBL height, and TKE values on (a) March 12, (b) June 10, and (c) August 22. The y-axis is the altitude above mean sea level (MSL).**

[Figure]

*date, time, alt*

*pres_ens_mean ?*

285

[Figure]

**Figure 6. ARMTRAJ-AAF 5-day back trajectory properties on March 12 (a) and (b), June 10 (c) and (d), and August 23  (e) and (f). The illustrated ensemble mean trajectories are calculated based on the flight coordinates and altitude range during measurement periods in (a), (c), and (e) and based on the ground level at the ArcticShark's flight coordinates in (b), (d), and (f). The star is the central facility at the SGP site. All figures are generated using Natural Earth by MATLAB®.**

*New Section*

The back trajectory of airmasses can support aerosol studies by providing context to long-range aerosol transport and suggest potential interactions during their path. Figure 6 presents a comparison of ARMTRAJ 5-day back-trajectory properties on three separate dates: March 12, June 10, and August 23 *2023*. For each date, two types of trajectories are shown: one set based on flight coordinates and altitude range during measurement periods (Figures 6(a), 6(c), and 6(e)) and another set of trajectories initialized at the ground level at the ArcticShark's flight coordinates (Figures 6(b), 6(d), and 6(f)). The ensemble statistics presented here are based on 25-member ensembles generated for each trajectory initialization altitude over a 5x5 grid typically spanning several kilometers to each direction relative to the ArcticShark's coordinates. The ensemble mean trajectories calculated using the flight altitudes (Figures 6(a), 6(c), and 6(e)) indicate longer travel pathway distances compared to those calculated based on ground-level initialization. The trajectories suggest that the flight period measurements included a wider range of atmospheric conditions and altitudes, capturing more variant and extended airmass pathways compared to ground-level measurements that were more localized. On March 12, The airmass trajectories originated mainly from the north region of the US before reaching the sampling area near the Southern Great Plains (SGP) site. The trajectories showed that the airmass traveled from the southwest on June 12. On August 23, the airmass was more influenced by the southeast region. The differences in airmass origin and trajectory paths between the three dates could be attributed to seasonal atmospheric circulation patterns, which vary with changes in temperature, pressure systems, and overall weather conditions.

**Table 2. Chemical composition of UAS filter samples measured by the MN-AMS analysis**

| Start time | End time | Ambient Mass concentration ($\mu g/m^3$) | Volume fraction | | | O/C | H/C | Organic density calculated ($kg/m^3$) | $\kappa_{Org}$ | $\kappa_{Overall}$ |
|---|---|---|---|---|---|---|---|---|---|---|
| | | | $(NH_4)_2SO_4$ | $NH_4NO_3$ | Organics | | | | | |
| 3/9/23 21:13 | 3/9/23 23:04 | 6.0 | 0.06 | 0.19 | 0.75 | 0.3459 | 1.6817 | 1141 | 0.10 | 0.24 |
| 3/10/23 16:43 | 3/10/23 21:15 | 5.3 | 0.10 | 0.32 | 0.58 | 0.4758 | 1.6038 | 1249 | 0.20 | 0.39 |
| 3/12/23 16:20 | 3/12/23 20:27 | 3.2 | 0.06 | 0.26 | 0.68 | 0.3358 | 1.6897 | 1132 | 0.09 | 0.27 |
| 3/13/23 15:26 | 3/13/23 18:01 | 3.5 | 0.07 | 0.13 | 0.80 | 0.3061 | 1.7107 | 1106 | 0.06 | 0.18 |

[Figure]

| | | | | | | | | | | |
|---|---|---|---|---|---|---|---|---|---|---|
| 3/14/23 17:25 | 3/14/23 21:28 | 3.2 | 0.08 | 0.19 | 0.74 | 0.3555 | 1.6626 | 1153 | 0.11 | 0.25 |
| 3/17/23 21:30 | 3/17/23 23:39 | 4.3 | 0.08 | 0.03 | 0.89 | 0.3074 | 1.7134 | 1106 | 0.07 | 0.13 |
| 6/8/23 16:28 | 6/8/23 19:07 | 5.2 | 0.10 | 0.02 | 0.88 | 0.4669 | 1.6102 | 1241 | 0.20 | 0.25 |
| 6/9/23 13:49 | 6/9/23 17:40 | 5.9 | 0.08 | 0.03 | 0.90 | 0.4310 | 1.6497 | 1206 | 0.17 | 0.22 |
| 6/10/23 21:00 | 6/11/23 1:06 | 27 | 0.05 | 0.01 | 0.93 | 0.5367 | 1.6438 | 1274 | 0.26 | 0.28 |
| 6/12/23 17:45 | 6/12/23 19:42 | 6.9 | 0.05 | 0.02 | 0.93 | 0.4019 | 1.6849 | 1177 | 0.14 | 0.18 |
| 6/22/23 14:13 | 6/22/23 16:58 | 8.5 | 0.07 | 0.02 | 0.91 | 0.4624 | 1.6463 | 1227 | 0.19 | 0.23 |
| 6/23/23 14:32 | 6/23/23 16:13 | 8.7 | 0.09 | 0.03 | 0.89 | 0.4279 | 1.6697 | 1198 | 0.17 | 0.22 |
| 8/17/23 14:58 | 8/17/23 17:40 | 8.8 | 0.20 | 0.07 | 0.73 | 0.3316 | 1.8754 | 1080 | 0.09 | 0.23 |
| 8/18/23 14:04 | 8/18/23 17:44 | 7.6 | 0.10 | 0.04 | 0.87 | 0.3330 | 1.8764 | 1081 | 0.09 | 0.16 |
| 8/21/23 14:45 | 8/21/23 17:08 | 14 | 0.12 | 0.04 | 0.84 | 0.2991 | 1.9034 | 1052 | 0.06 | 0.15 |
| 8/22/23 14:45 | 8/22/23 17:56 | 8.9 | 0.09 | 0.03 | 0.87 | 0.3217 | 1.8876 | 1071 | 0.08 | 0.15 |
| 8/23/23 14:14 | 8/23/23 17:39 | 14 | 0.07 | 0.03 | 0.90 | 0.3109 | 1.9047 | 1060 | 0.07 | 0.12 |
| 8/24/23 13:40 | 8/24/23 17:43 | 7.5 | 0.09 | 0.03 | 0.88 | 0.3064 | 1.9069 | 1056 | 0.06 | 0.13 |
| 8/26/23 13:51 | 8/26/23 19:38 | 10 | 0.08 | 0.03 | 0.89 | 0.2910 | 1.9073 | 1046 | 0.05 | 0.11 |
| 8/27/23 13:58 | 8/27/23 17:14 | 8.8 | 0.12 | 0.04 | 0.83 | 0.3100 | 1.8935 | 1062 | 0.07 | 0.16 |

[Figure]

| 8/29/23 15:18 | 8/29/23 19:49 | 39 | 0.13 | 0.05 | 0.82 | 0.4321 | 1.8664 | 1146 | 0.17 | 0.25 |
| 8/30/23 15:17 | 8/30/23 20:32 | 28 | 0.29 | 0.11 | 0.60 | 0.4766 | 1.7565 | 1204 | 0.21 | 0.37 |

**3.2 Case study with unique measurement capabilities**

*So not all flights had chem. analysis?*

[revised manuscript text omitted]

---

## Author Comment (AC1)

*The paper summarizes measurements made onboard the Arctic Shark above the SGP ARM site over a period of 3 years. The paper achieves its stated goal of demonstrating the use of the UAS for observations of meteorological and aerosol properties. The following comments should be addressed before publication.*

Response: Thank you very much for your constructive and helpful review comments. We appreciate your time and effort in providing detailed feedback on our work. Your insights have been invaluable and significantly contribute to improving the quality and clarity of our manuscript. We have drafted the responses below and revised the manuscript accordingly.

*Table 1: It would be helpful to add the actual instrument and manufacturer information to the table.*

Response: Thank you very much for your great suggestions. Table 1 aims to be in accordance with the ACP data policy. We have included two more tables in the supplemental file and added the information in the manuscript, "More information about the instrumentation has been published before and included in the supplementary (Mei et al., 2022; Mei et al., 2024)."

*Measured particle size ranges should be provided for all in situ particle measurements.*

Response: we added the size ranges in the supplemental file and the manuscript. For example, the Figure 2 caption has been revised to "Atmospheric conditions encountered during the March, June, and August 2023 flights. (a) ambient temperature; (b) ambient relative humidity; (c) total number concentration from the mixing condensation particle counter (CPC, > 7 nm); and (d) total number concentration from the portable optical particle spectrometer (POPS, 135 -3,000 nm)."

*Lines 122 – 123: What exactly is the ArcticShark chemical filter collecter? What filters are used? What is the size range?*

Response: we added, "ArcticShark is equipped with an eight-spot filter sampler (Model 9401, Brechtel), which collects ambient particles at a 2.5 lpm flow rate on the 13 mm polytetrafluoroethylene (PTFE) filter media. " "The chemical compositions of collected samples from 2023 deployments were included in Table 2 and discussed in section 3.1.5. "

*Lines 156 – 158: How does the assumption of a uniform aerosol composition affect the removal of the influence of humidification from the extinction profiles and retrieval of the vertical CCN concentrations? The assumption is expected to have an impact on the accuracy of the retrievals for dust versus sulfate, for example.*

Response: Great point. We agreed that the current assumption would affect the accuracy of the retrieved vertical CCN concentrations. We have another airborne study using the piloted aircraft data to investigate this issue. Although the other manuscript is still under review, the analysis showed a moderate correlation between dry-corrected extinction and airborne CCN data after assessing the data between the vertical CCN concentrations obtained from extinction-, satellite-, and model-based retrieval methods and airborne CCN concentrations collected at 0.24% supersaturation (SS) within the 3, 9, 27, and 81 km

regions centered over the SGP site during the spring and summer of 2016. We found the retrieved number concentration of CCN (RNCCN) method showed regression best-fit slopes close to unity and consistent prediction errors for most of the data. Due to the limited scope of this measurement report, we have revised and added the information below in the manuscript, "Note that the assumption of uniform aerosol composition in the current VAP increases the uncertainty of the vertical CCN concentration retrievals. "

*Figure 2d: please provide the particle size range measured by the POPS.*

Response: the Figure 2 caption has been revised to "Atmospheric conditions encountered during the March, June, and August 2023 flights. (a) ambient temperature; (b) ambient relative humidity; (c) total number concentration from the mixing condensation particle counter (CPC, > 7 nm); and (d) total number concentration from the portable optical particle spectrometer (POPS, 135 -3,000 nm)."

*Line 202: Please provide more details about the "haze environment" in August. Is this due to agricultural burning?*

Response: we added, "we observed a notable increase in particle concentration close to the ground in August, which might be related to the haze environment prevalent during that month and due to local agricultural burning events. "

*Line 207: Please provide the size range of the accumulation mode that is referred to here.*

Response: added.

*Line 208: With the chemical composition information couldn't more said here about the importance of secondary particle formations and emissions from agricultural sources?*

Response: We acknowledge the importance of further exploring secondary aerosol formations and local emissions. Our collaborators are actively working on a detailed analysis of the collected samples, with results to be presented in a future publication. However, given the focused scope of this ACP measurement report, we have limited our discussion to findings related to vertically resolved atmospheric properties.

*Figure 3 a and b: Any explanation for the offset between the AIMMS-30 and LICOR aboard the UAS?*

Response: we added, "The discrepancy between the AIMMS-30 and LiCor measurements can be attributed to several factors, including the spatial separation of the two platforms and the performance degradation of the AIMMS-30 sensor."

*Figure 3 c and d: Coefficients of determination show good agreement between the UAS and balloon wind direction and speed but there is a lot of spread in the data. Any explanation? How far apart were the two platforms?*

Response: we added, "The radiosonde provided a snapshot of wind conditions as it ascended over time, while the UAS profiling above the SGP site captured more measurements within the same altitude range, covering a much larger spatial area. Additionally, the spatial separation (up to 6 km) between the two platforms also contributes to the scattering of comparison."

*Figure 4: How is the PBL height determined?*

Response:

We used the best estimate PBL height derived from multiple lidar-based PBL height estimates and ancillary environmental parameters (Zhang et al., 2025). In section 2.3, we have added in line 166-170, "For boundary layer (PBL) height estimations, we overlayed our flight tracks with the best estimate PBL height derived from multiple lidar-based PBL height estimates and ancillary environmental parameters (Zhang et al., 2025). The multiple lidar-based PBL height estimates include PBL height from ceilometer (CEILPBLHT) (Zhang et al., 2022), from Micropulse Lidar (PBLHTMPL1SAWYERLI), from Doppler Lidar (PBLHTDL), and from Raman Lidar data (PBLHTRL1ZHANG) VAPs.." In section 3.1.,4, We referred to section 2.3 for PBL height estimates.

Reference:

Zhang, D., Comstock, J., Sivaraman, C., Mo, K., Krishnamurthy, R., Tian, J., Su, T., Li, Z., and Roldán-Henao, R.: Best Estimate of the Planetary Boundary Layer Height from Multiple Remote Sensing Measurements. Submitted to AMT.

*Lines 250 – 252 are redundant with lines 224 – 226.There is a lot of repeated text between pages 11 and 12.*

Response: Thank you for catching that. We have removed the redundant materials.

*Line 264: Fight days in March? Should be flight days.*

Response: corrected.

*Table 2: are 4 to 5 significant figures warranted for reported O/C and H/C ratios?*

Response: Thank you for pointing it out. 3 significant figures are more reasonable with the AMS analysis. We have revised the table.